# The More Rural the Less Educated? An Analysis of National Policy Strategies for Enhancing Young Adults' Participation in Formal and Informal Training in European Rural Areas

Julia Weiss [1,*] and Christin Heinz-Fischer [2]

1   GESIS Leibniz Institute for the Social Sciences, 68159 Mannheim, Germany
2   Institute for Political Science, University of Heidelberg, 69115 Heidelberg, Germany
*   Correspondence: julia.weiss@gesis.org

**Abstract:** Young adults in rural areas in many cases have limited educational opportunities. To obtain higher educational qualifications, rural youth often travel long distances. Therefore, many rural youths choose the "shorter" route and complete vocational training closer to home or drop out of their education prematurely. Against this background, this study examines the education policies of European countries and explores the extent to which these problems are addressed within their policy framework and what measures are taken to improve the situation. Using a unique dataset of policies of 31 European countries for the period 2010 to 2020, we examine more than 500 national and subnational policies that address formal and informal education and training. The results show that despite the sometimes high number of rural youths not in education, employment, or training (NEETs) and high early-school-leaving rates, only some countries have developed a respective policy strategy. The analysis presents the different measures implemented in the countries and furthermore shows that a high problem pressure in a country is not necessarily accompanied by a consideration in policy. Prospectively, there is a need for new policies that understand the multidimensionality of the issue and significantly improve the situation of rural youth.

**Keywords:** rural; youth; education; training; policy; Europe; NEETs





## 1. Introduction

The problem of youth unemployment has long been a high-profile issue on the agenda of EU policymakers and has resulted in major policy instruments such as the Youth Guarantee [1]. At the same time, the issue of the education of young adults has lagged behind in relevance, although it is clear that the pathway to employment starts with education and studies have shown that educational attainment is a strong predictor of future labor market outcomes [2]. As part of the Europe 2020 strategy, the EU aimed to reduce the rate of early school leavers to 10 percent and to increase the share of people with tertiary education to 40 percent by 2020. To achieve this ambitious goal, actions across all domains, including all geographical areas of the EU member states, are needed.

However, so far it seems as if rural youth are of minor importance in EU policies. The EU 2020 strategy, for example, refers to rural areas, and their development, and youth in general, but never specifically to the situation of youth in rural areas. Youth in rural areas themselves feel that they are provided with fewer opportunities compared to urban youth and that living in rural areas makes it harder to realize their goals and ambitions [3].

Looking at rural youth and how they are represented in policies matters because young adults are essential to the future development and sustainability of rural areas [1]. Around 75% (2015) of the EU territory consists of rural areas with around 29% (2018) of the population living there [4,5]. The situation in rural areas is constantly changing, be it, for example, due to changes in the agricultural sector, which is the driver for rural development in large parts of the EU, or changes in daily mobility patterns due to the

higher availability of jobs in urban areas, thus influencing the life and work of those living in rural areas [6]. Furthermore, the living conditions often differ greatly from those in other, more urban, regions. For instance, the educational level of rural youth over the past ten years has always been below that of youth in cities, towns, and suburbs [7].

As previously mentioned, goals with regard to education and early school leaving are set up within the framework of the open method of coordination (OMC) at the EU level; concrete policies need to be implemented by national governments (or, depending on the structure in the country, at the regional level). Thus, EU member states implemented several policies, for example, to prevent early school leaving [8]. All acknowledge that there are manifold reasons for young adults to leave school, which besides personal needs are associated with mainstream education systems [8]. The composition of educational systems sometimes prevents successful educational paths, which applies all the more to rural areas.

So far, the state of research lacks a systematic analysis of education policies with a focus on rural youth in European countries. Instead, topics such as lifelong learning have received attention [9,10] or processes such as the Bologna process have triggered analyses of higher education policies [11,12]. With the aim of filling this research gap, this article offers an overview of which states address the topic within their policies and how they want to improve the education of rural youth. For this goal, we generated a new dataset regarding the role of education of rural youth in national policies (or, where applicable, subnational policies). This dataset extends beyond research investigating general policies with regard to youth and is unique according to our knowledge.

The remainder of this study is structured as follows. The subsequent section gives an overview of the situation of rural youth in the past decade in terms of their proportion of the population, their level of education and the proportion of early school leavers. Subsequently, a brief look at the current state of research is given before the data collection and methodological approach are described in the fourth section. This is followed by the presentation of the analysis. Finally, results are summarized and discussed.

## 2. What Is the Situation of Rural Youth in European Countries?

The number of young adults in rural areas varies greatly between European countries. Regarding the definition of "rural" and "youth", we use the empirical definitions of the EU itself, i.e., according to Eurostat (2018) a rural area is "an area where more than 50% of its population lives in rural grid cells" [13]. For youth, there is no clear definition since the legal thresholds differ between the EU member states [14]. Therefore, based on the data availability, we define youth as 15–24-year-olds. Parts of the data are only available for subgroups of this age range.

In 2019, the countries with the highest share of rural youth relative to their total youth population were Slovakia (46.3pp), Romania (45.7pp), Slovenia (44.9pp), and Lithuania (44.5pp) (see Figure 1). From 2010 to 2019, the share of rural youth (relative to the total youth population of the country) increased particularly strongly in Luxemburg (+19.70%) and France (+16.43%). At the same time, it decreased strongly in Sweden (−36.93%) and Finland (−31.71%).

The share of early school leavers also varies widely across countries (see Figure 2) and does not necessarily correlate with high shares of rural youth. Iceland (29.4pp), Bulgaria (24.5pp), and Romania (22.4pp) make up the group of countries with the highest early-school-leaving rates in 2019. The lowest rate is found in Switzerland with 2,4pp. From 2010 to 2019, Czechia (+30.6%), Hungary (+26.57%), Sweden (+25.37%), and Slovakia (+21.54%) were the countries with the highest increase in early-school-leaving rates. With the exception of Hungary, however, these countries remained below the 10% mark. From 2010 to 2019, a different group of states experienced a sharp decline in their early-school-leaving rates. These include Greece (−63.5%), Cyprus (−62.9%), Ireland (−59.09%), Portugal (−58.36%), Lithuania (−57.7%), Luxembourg (−57.01%), and Belgium (−53.33%). With the exception of Portugal, all countries in this group made it below the 10% mark during this

period. Overall, in 2019, 19 of the 31 states achieved the EU target of an early-school-leaving rate below 10%.

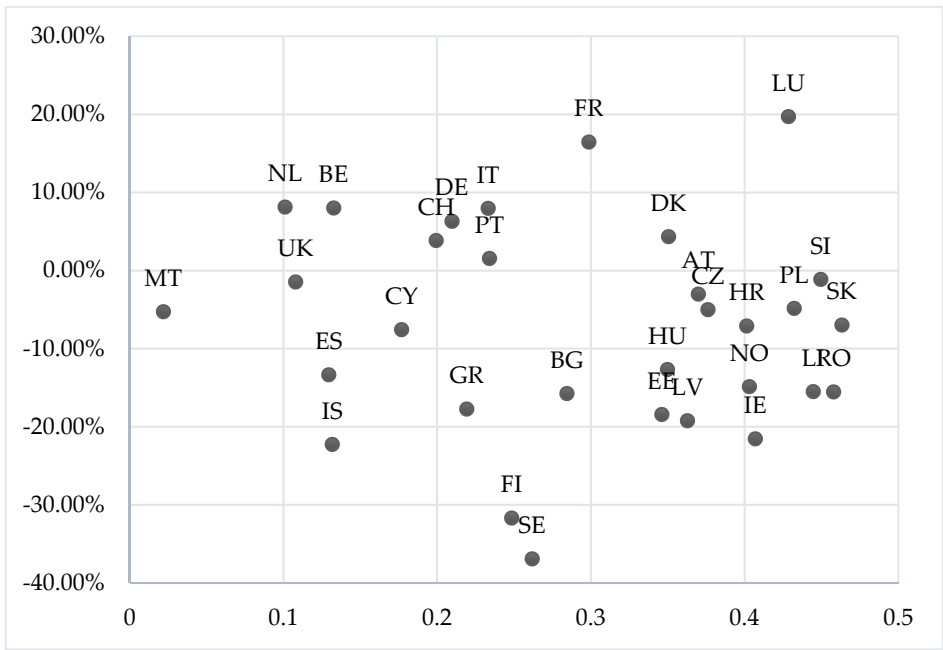

**Figure 1.** Percentage points of rural youth population relative to total youth population (15–24 years) in 2019 and relative change in percentage points 2010–2019 in percent. Source: Eurostat 2020 [lfst_r_pgauwsc].

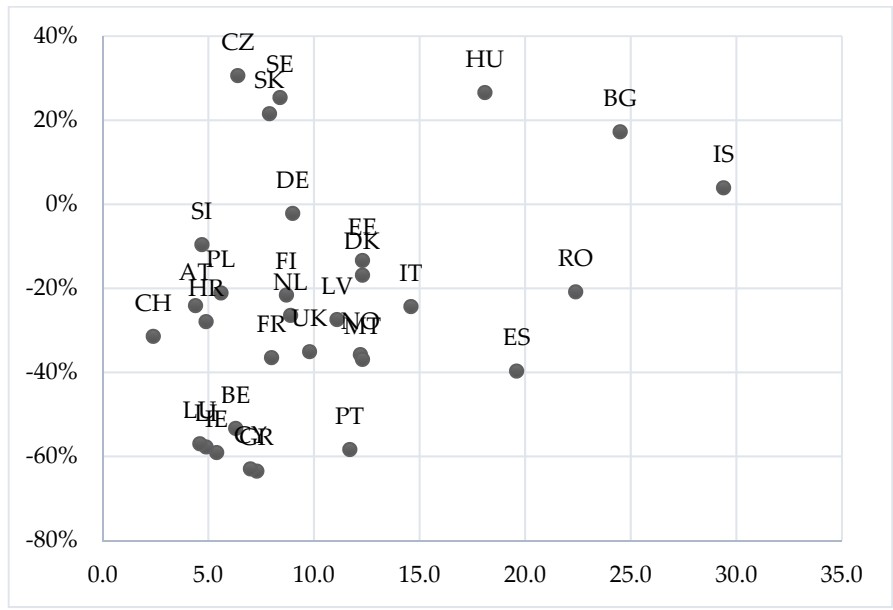

**Figure 2.** Percentages points of early school leavers (18 to 24 years) in rural areas in 2019 and relative change in percentage points 2010–2019 in percent. Source: Eurostat 2020, [edat_lfse_30].

The EU target for tertiary education is missed by all European countries (see Figure 3). In all countries, the lowest proportion of rural youth achieved tertiary education. The highest proportion of tertiary educated rural youth can be found in Cyprus, the lowest in Denmark. In more than half of the countries, however, the majority of rural youth only achieved a low level of education (ISCED 0–2).

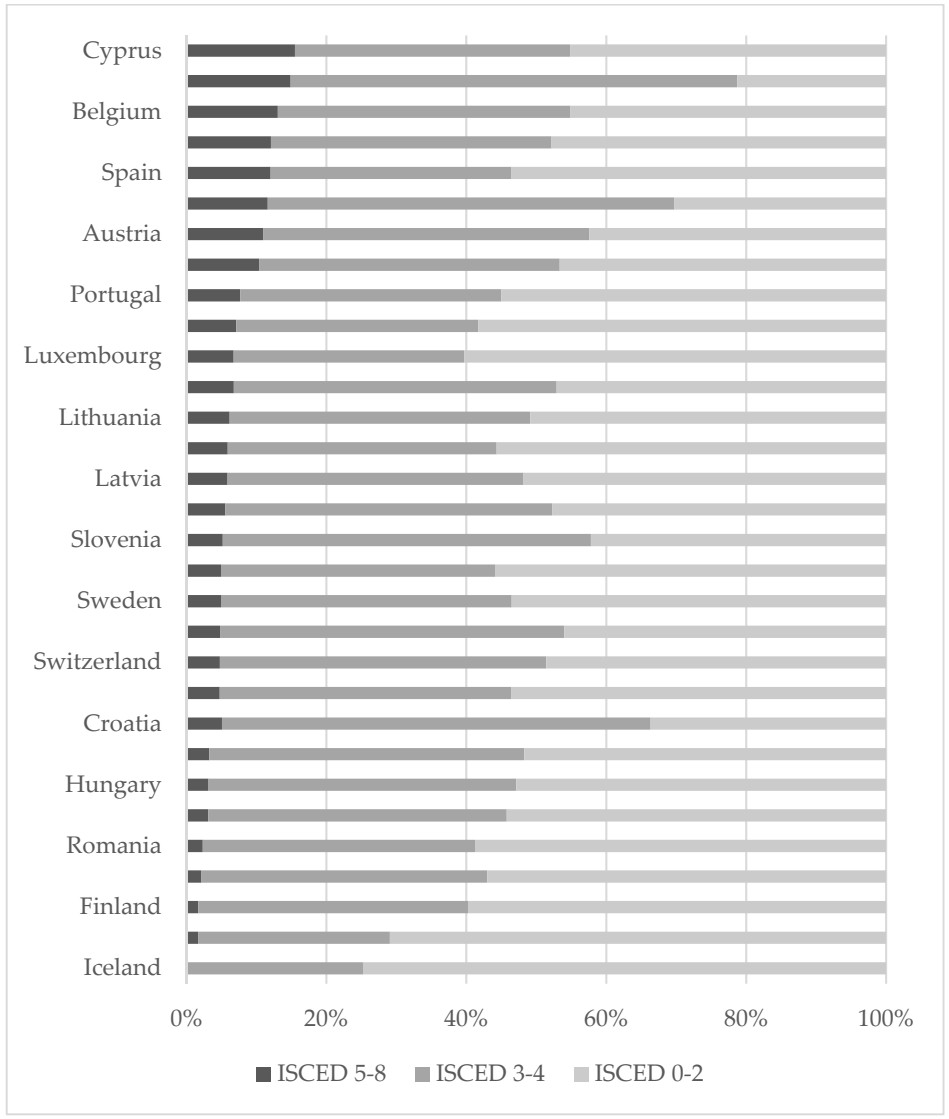

**Figure 3.** Educational attainment of rural youth, 15–24 years (average 2010–2019, percentages).Data for tertiary education for 15–24-year-olds in rural Iceland are not available (confidential) in the Eurostat data (hence the 0% ). There are data for 30–34-year-olds in rural areas: 36.9% had higher education (average for 2010–2019), but this rate is significantly lower than the average for the urban population (54.9%). In Iceland as a whole, a very small proportion (between 1.5 and 3.8%) of younger people (15–24-year-olds) have completed tertiary education, because entry into tertiary education in Iceland generally occurs later than in many countries (OECD 2014). Source: Eurostat 2020, [edat_lfs_9913].

Overall, it becomes clear that many young people live in rural areas and that there is a great need for action to improve their situation. The early-school-leaving rate is above the 10 percent mark in roughly one-third of the countries and rural youth are often only poorly educated.

## 3. State of Research—Education of Young Adults in Rural Areas

Previous studies offer a wide range of reasons that make educational policy for rural youth an important research topic. For example, they show that rural youth due to their geographical location achieve different educational outcomes than their counterparts in cities, towns, and suburbs. First, the local job market, mainly characterized by agriculture, manufacturing, or service, often does not require higher education credentials [15]. Second, postsecondary education often requires moving away from home, since respective educa-

tional institutions are rare in rural areas. Young adults often do not want to abandon their connections to family and community by moving and therefore lower their educational aspirations [16,17].

Previous studies further showed that schools positively shape the educational outcomes of rural youth [15]. With this regard, not only structural characteristics of schools are important but also subjective experiences made in school [18].

Schools in rural areas themselves are often faced with problems such as difficulties in recruiting highly qualified teachers or inadequate infrastructure. This is often due to the small size of rural communities, which makes it much more expensive per capita for them compared to urban areas and implies high fixed costs to maintain small schools [19]. Seen as a whole, "a lack of a critical mass of students (and teachers and other staff) together with limited budgets and sparse populations often means that many rural families have a limited choice of schools, education programs, after-school activities, and access to additional support" [19].

Studies show that it is important to address the differences in educational opportunities between different urban regions, as they show that school systems that have been successful in closing the rural–urban gap show higher academic performance [19]. Moreover, poor educational opportunities in rural regions have negative consequences for the regions themselves. Education can reinforce patterns of rural out-migration since out-migrants from rural areas are usually young and better educated or looking for better education [20]. Consequently, processes of local economic downturn and disadvantage may worsen "as a matter of migration stream selectivity" [21]. While a lot of research has been conducted on individual conditions and socio-economic context factors, only a handful of studies have looked at actual policy strategies. Some analyze the United States' education system and criticize "one-size-fits-all approaches" which do not take the differences of rural schools into consideration and might even affect these schools and communities negatively, consequently holding back rural development [21,22]. For the European context Gristy et al. [23] show that political legacies such as liberal democracies versus post-communist states have an impact on the education policies and patterns influencing rural education such as school closures.

Looking at the landscape of education policies in the EU, it must always be kept in mind that education is not a genuine area of competence of the EU but is addressed within the OMC framework. The OMC, as a form of policy learning [24,25], targets policy learning as "a relative enduring change in behavior that results from experience" [26]. This learning process is circular, starting with recommendations from the EU, continuing with implementation in the respective country, and ending with a subsequent evaluation of the results achieved, which is then carried out again by the EU. Even if best practice examples from other countries are often consulted, especially with regard to implementation, the specification of targets by the EU remains a central component of this process. The extent to which the EU's targets, for example in relation to the proportion of youth which are not in education, employment, or training (NEETs), have had an influence on policymaking with regard to the education of rural young people will be considered in the following. Thereby and with this systematic overview, we contribute to the literature on European rural youth education policies.

## 4. Data Basis and Analysis Procedure

The observation period is 2010–2020, which corresponds to the EU 2020 strategy period, and the sample consists of 31 European countries. To create the database, we identified three types of (sub-)national policies in which the education of rural youth could potentially be addressed. These are national education strategies [27], national youth strategies [28] and rural development programs [29]. For a long time, the nation state was the dominant level when it came to the policy field of education, but in recent years this has changed somewhat. In the meantime, the EU itself has set the tone for education policy and, within the framework of the OMC, there is also an exchange of best practice

between the nation states [30]. However, European influence does not take place equally in all areas of education. Instead, it is much more pronounced in the area of tertiary education, for example the Bologna process or the Erasmus program, than in other areas of education. Even if the EU does not make this mandatory, European influences are nevertheless reflected in national education strategies (NESs). Similar observations can be made in the course of national youth strategies (NYSs). Here, too, European influence is increasing in a genuinely nation-state policy field. This is a development that has already been described above, particularly in the context of combating youth unemployment in the recent past. In comparison, Regional Development Programs (RDPs) are the main instrument of the second pillar of the EU's common agricultural policy and have to be planned and implemented by all member states since their introduction in the context of the Agenda 2000 reform [31]. The multi-annual programs are defined and implemented by the member states at national or regional level. At the end of their term, they are then subjected to an evaluation process by the EU.

Based on information of the EU education information network "Eurydice", and OECD publications on education policies, we categorized the states according to their degree of centralization with regard to responsibilities in education policy. For centralized states (e.g., France and Greece), we examined national policies, and for decentralized states (e.g., Germany and Czechia), we additionally analyzed policies at the respective regional levels. This strategy is particularly suitable, since previous studies showed that governance in education has become increasingly decentralized [32,33].

In cases where we could not find the relevant documents via either EU, UNESCO (Planipolis), OECD (Education Policy Outlook Reforms Finder) or national ministry home-pages, we searched for them manually. In cases in which the documents were not available in English, we analyzed the documents in the local language with the help of the standard translation software Google Translate. We checked important sections of the documents with multiple translation tools to ensure a correct translation.

We coded all policies manually by using the keywords 'rural youth' and 'education' as well as variations of these terms. The coding scheme was developed in an explorative manner, applying the method of inductive content analysis: the first fifty policies were coded with a preliminary coding system, which was then changed according to the content of the policies. All texts of the pilot coding stage were then coded again. In the final coding scheme policies were coded as appropriate (=1) when the education of youth in rural areas was explicitly mentioned. Policies were coded as not applicable (=0) if non-specific measures were mentioned or if no mention or reference to the education of young adults in rural areas was made. This resulted in the coverage of 563 policies. For each of these 563 policies, we further documented summaries of the most important goals, relevant passages, information on the occasion of the creation of the policy, and the target group, as well as the targeted educational level, in an Excel spreadsheet. As will be shown below, 117 of these policies directly addressed the education of rural youth. These policies were then the subject of further content analysis. For this purpose, we analyzed the documents in more detail and in the context of the entire document, with regard to goals and measures. The results will be presented in the next section.

The second step of the analysis consists of a consideration of what factors promote a consideration of rural youth education in the policies. A detailed presentation of the sources and summary statistics of the quantitative analysis can be found in the Appendix A. The mentioning of rural youth education within a policy (yes/no) as a binary variable forms the dependent variable in the logistic regression. Regarding factors that possibly promote a consideration of rural youth within the policies, we focus on aspects that represent the thresholds of the EU within the EU 2020 strategy. These aspects are operationalized via the share of rural youth, the share of rural NEETs, the share of rural early school leavers as well as the educational level of rural youth. Here, we expect that with a high proportion of rural youth, rural NEETs, and rural early school leavers, there will be an increasing likelihood of consideration. The simple proportion of rural youth expresses the relevant target group,

whereby it is to be expected that the larger the target group, the more likely consideration will take place. With regard to rural NEETs and early school leavers, it has already been explained that certain targets are set by the EU, which means that it is to be expected that if these targets are not met, there is an increased likelihood that policy-makers will be encouraged to address the issue in their specific policies. Similarly, but in the opposite direction, this is to be expected with regard to the level of education. Based on the target value of the EU, we expect that an increased probability of being mentioned in the policies accompanies a decreasing educational level of the young adults in rural areas.

## 5. Results: Goals and Measures of Policies Addressing Rural Youth Education

Nineteen of the 31 states address the education of young adults in rural areas in one of their national policies (see Table 1). An overview of the percentage of policies in which the education of rural youth is mentioned as a proportion of the total number of policies considered per country can be found in Appendix A.

Most often, the issue is addressed in the national education strategies (NESs), followed by the regional development programs (RDPs) and the national youth strategies (NYSs). Of the twelve states that do not address the issue at the national level, five states address it at some regional level instead. These states are Belgium, Estonia, Sweden, Slovenia, and Slovakia. In the case of Estonia, at least half of the existing regions focus on the education of rural youth, while in the other four countries it is only considered in one or very few regions, like for example in Sweden, where only one region has a focus on rural youth education. Finally, seven countries do not address the education of rural youth at all. These are Greece, Hungary, Luxembourg, Malta, the Netherlands, Portugal, and Switzerland.

### 5.1. National Education Strategies

Of all national education strategies analyzed, twenty-one refer directly to the education of rural youth. Some start by describing the problematic educational situation in rural areas. The NES of Czechia (2020), for example, presents regional disparities in the quality of education, as well as socio-economic reasons for segregation tendencies in education.

"The Czech Republic suffers from great differences in the quality of schools. Surveys of inter-regional differences in the quality of education show that pupils living in structurally disadvantaged regions with a higher number of socially excluded localities score lower in tests verifying educational outcomes than pupils living in regions with a higher quality of life. Based on the results of international and national surveys, it can be stated that the achieved level of education in the Czech Republic is influenced not only by the socio-economic status, but also by the place of residence." (Czechia, NES 2020, translated)

Besides such descriptions, many of the strategies also contained concrete measured for improvement. Some countries provided direct financial aid. These were used, for example, for the implementation of adult education in rural areas (Croatia, NES 2014). Additional funding of small schools and promotion of cooperation between schools was a further important measure (Finland NES, 2012; UK, NES 2016 and 2017). The aim was often to prevent school closings in rural areas (Ireland, NES 2016 and 2019).

Other countries planned and implemented further concrete measures. A central theme here was spatial access to education. Often, access to higher education for rural youth is associated with mobility, as higher schools are located in suburban and urban areas. For this reason, Poland (NES, 2013) implemented a new concept for distance learning and Spain (NES, 2019) expanded non-formal and modular training, while Denmark (NES, 2018) created opportunities for boarding school access for rural youth in the context of vocational education. The UK took a different approach and implemented a new framework for university acceptance schemes to widen participation of students from all backgrounds (UK, NES 2011).

**Table 1.** Overview of educational policies (2010–2020).

| Country | Mention of Rural Youth Education within National Policy Level | | | | Mention of Rural Youth Education within Subnational Policy Level | | | Total Number of Examined Policies within Country | Centralization of Responsibility of Educational Policy on the National Level |
|---|---|---|---|---|---|---|---|---|---|
| | National Education Strategy | National Youth Strategy | Rural Development Program | Other | Subnational Education Strategy | Subnational Youth Strategy | Subnational (Rural) Development Strategy | | |
| Austria | 1 | | 1 | | | | | 6 | centralized |
| Belgium | | | | | | | 1 | 11 | |
| Bulgaria | | 1 | 1 | | | | 8 | 33 | |
| Croatia | 1 | | 1 | | | | | 3 | centralized |
| Cyprus | | | 1 | | | | | 4 | centralized |
| Czechia | 1 | | | | | | | 19 | |
| Denmark | 1 | | 1 | | | | 1 | 11 | |
| Estonia | | | | | | | 7 | 18 | |
| Finland | 3 | 1 | | | 1 | | 3 | 30 | |
| France | | 1 | | 1 | | | | 6 | centralized |
| Germany | | | 2 | 1 | 15 | | | 41 | |
| Greece | | | | | | | | 3 | centralized |
| Hungary | | | | | | | | 2 | centralized |
| Iceland | | | 1 | | 2 | | 1 | 51 | |
| Ireland | 3 | | | | 3 | | | 23 | |
| Italy | 1 | | | 1 | 2 | | 1 | 23 | |
| Latvia | 2 | | | | | | 3 | 9 | |
| Lithuania | 1 | | 1 | | | 1 | 15 | 63 | |
| Luxembourg | | | | | | | | 5 | centralized |
| Malta | | | | | | | | 4 | centralized |
| Netherlands | | | | | | | | 4 | centralized |
| Norway | | | 1 | | | | 1 | 17 | |
| Poland | 1 | | | | 1 | | 5 | 19 | |
| Portugal | | | | | | | | 4 | centralized |
| Romania | 2 | 1 | 1 | | | | | 4 | centralized |
| Slovakia | | | | | 1 | | | 11 | |
| Slovenia | | | | | | | 3 | 16 | |
| Spain | 1 | | | 1 | 1 | | | 22 | |
| Sweden | | | | | | | 1 | 26 | |
| Switzerland | | | | | | | | 22 | |
| United Kingdom | 3 | | | | | | | 13 | |
| Total | 21 | 4 | 11 | 4 | 26 | 1 | 50 | 523 | |

Another important aspect was the equipment and role of schools. Some countries mentioned the availability of qualified teachers as a central problem. To counteract the shortage, Italy (NES, 2015) renewed the concept of distribution of staff between the regions and Romania (NES, 2015) developed new approaches to ensure the provision of a sufficient number of qualified teachers. Further, some states focused on the social and cultural role of schools in rural areas.

"The goal is to develop operating models of ( . . . ) small rural schools as multifunctional centers, taking into account the factors influencing the operation of the school, the educational, social, cultural and economic needs of local communities ( . . . )." (Latvia, NES 2014, translated)

Here it was about seeing schools as multifunctional local community centers and equipping them accordingly (Latvia, NES 2012 and 2014). Lithuania emphasized the need to take the composition of the rural population into account and, for example, offered education in the languages of national minorities (Lithuania, NES 2013). Thus, schools were seen as networks of rural areas that help to reduce social exclusion.

Finally, the analysis of the policies enables us to examine the occasions of state policymaking with regard to education of rural youth. The vast majority mentioned their policymaking as being initialized by the EU 2020 strategy. Another reason, at least for Denmark, Finland, and Ireland, was changing labor market structures and the demand for skilled workers. In other countries, the issue of the education of rural youth was a focus of the national governments and finally, in the case of the UK, the results of the Pisa studies and the sustainable development goals gave rise to policy development.

### 5.2. National Youth Strategies

The education of rural youth is not primarily a topic in national youth strategies and yet four countries, Bulgaria, Finland, France, and Romania, address it in this context. Their measures overlap greatly with those other countries implemented in the context of their national education strategies. A central topic, again, is the effective access to education for rural youth. In Bulgaria (NYS, 2010) this was pushed through the promotion of the EU Youth in Action Program, while Finland (NYS, 2012) established an educational equality program. Romania (NYS, 2015) focused on the access to tertiary education by providing mechanisms for financing tertiary education for young people from disadvantaged rural areas. The issue of the number of teachers played a role in France.

"After the massive job cuts that have taken place in recent years, the creation of new jobs for school teachers from the start of the 2012 school year will allow for an initial strengthening of class supervision, particularly in schools which are faced with the most complex situations. These jobs will be mobilized to improve the reception of students and promote their success, in particular in priority education schools and in isolated rural areas." (France, NYS 2013, translated)

Further, France (NYS, 2013) strengthened a new aspect, by implementing local actions to promote access to quality educational leisure activities as an essential complement to school education for youth in distant areas.

In the case of national youth strategies, only Bulgaria mentioned the influence of the EU, as their policy strategy was a reaction to the recommendations of the Council of Europe. In the other three states, the government's focus initialized their policymaking.

### 5.3. Rural Development Programmes

Since the agricultural sector is the most strongly Europeanized policy area in the EU [34], it is not surprising that all the policies analyzed in the area of rural development programs mention the second pillar of the Common Agricultural Policy as the occasion for their policy. Being an EEA partner, this also applies to Norway (RDP, 2013). A central goal was pursued in this context:

"The central challenge for rural regions is to maintain or re-create high-quality, differentiated and easily accessible educational provision for the relevant age groups, in order to

improve the prospects of families and young people to stay and to meet the skilled labor needs of the future." (Germany, RDP 2020, translated)

To achieve the improvement of the educational situation of rural youth the states used different measures. Two main paths were taken here. First, several states implemented measures to improve training. Particular attention was paid to the development and improvement of vocational training (Austria, RDP 2014; Croatia, RDP 2015; Cyprus, RDP 2015; Denmark, RDP 2014). Depending on the country, vocational training for agriculture and forestry should be improved or introduced at all. Denmark, for example, mentioned a large number of concrete measures with regard to vocational training:

"The Danish business community generally needs the training of more skilled workers. Thus, there is a need to strengthen the search for vocational education. For the farmers, operations managers and forestry technicians, since 2005 a constant level has been found in the number who have completed the education. The general problems with non-application for vocational education are sought to be resolved through other national information measures. The regions, business schools, UU centers [municipal youth guidance centers] and the municipalities have in collaboration launched a number of initiatives to promote the search for vocational education and to ensure the retention of students in education. It is about improving the transition from primary school to vocational education and to higher education, special internship campaigns, better regional education coverage, strengthened professionalism, etc." (Denmark, RDP 2014, translated).

Second, states again chose financial support. This took place in the form of start-up aids for young farmers (Austria, RDP 2014; Cyprus, RDP 2015) or as support for municipal educational infrastructures (Bulgaria, RDP 2015; Romania, RDP 2015).

*5.4. Subnational Strategies*

The subnational strategies also refer to a vast majority of cases to the influence of the EU, either with respect to the second pillar of the agricultural policy or within the framework of the EU 2020 strategy, when it comes to the cause of the policy. With a look at the concrete measures and goals, three major themes emerge.

First, the (re)structuring of the schools: in rural areas, there are often only small elementary schools left, if any, whose existence is severely threatened. The regions in the states are taking very different approaches to dealing with this problem. There are two general options. Either the preservation of schools, for example by creating multi-grade classrooms (e.g., Germany: Brandenburg, Hesse) or special financial support for preservation (e.g., Italy: Liguria), or the abandonment of small schools in the course of forming school centers (e.g., Spain: Comunidad Foral de Navarra and Germany: Saxony-Anhalt). In the same context, some regions mention the need to strengthen or to implement vocational education training centers in rural areas (e.g., Bulgaria: Varna, Denmark: Nordjylland, Finland: Central Ostrobothnia, and Germany: North Rhine-Westphalia and Saxony).

Second, the policies at this level address changes in the forms of learning. In terms of diversification and adaptation to the needs of rural regions, e-learning as a form of distance learning is a major topic. Many regions are therefore planning to introduce or expand these forms of learning in rural areas (e.g., Finland: Lapland and Kainuu), Iceland: Blönduósbær, Ireland: Cork, Italy: Sicily, Norway: Innlandet County). Third, some regions choose to see schools in rural areas as small town centers and to build them up accordingly. Further forms of public and social services are integrated into the buildings or the schools are upgraded as a social core through events and further public functions (e.g., Estonia: Lääne-Viru, Iceland: Kaldrananeshreppur, Bulgaria: Shumen, Latvia: Latgale Region, Lithuania: Alytus District Municipality, Radviliškis District Municipality, Rokiškis District Municipality, Ukmergė District Municipality).

In addition, some regions are pursuing other strategies. Here, for example, young farmers and their (vocational) training are again in the foreground (Belgium/Dutch Community, SubRDP 2015). In addition to the establishment of vocational training, the Pomurska region in Slovenia also implemented life-long learning centers in rural areas (SubRDP,

2015). In some regions, incentives for teachers are being considered more closely (Estonia: Rapla, Germany: Mecklenburg-Western Pomerania) and, finally, in one region the topic of preschool education is being discussed (Lithuania: Mažeikiai District Municipality).

## 6. What Conditions the Observance of Rural Youth Education in Policies?

Besides the look at the content of the existing policies provided by the previous analysis, the question now arises, as to which factors promote mentioning/non-mentioning rural youth education in policies. Here, the factors that were used above to describe the situation of rural youth are now brought together with the probability of taking into account rural youth education in policies.

Needs thus build the basic demand structure. Therefore, the greater the proportion of young adults in rural areas, the proportion of rural NEETs and the proportion of rural early school leavers and the smaller the share of rural youth with tertiary education, the greater should be the likelihood of mentioning rural youth education in policies. This is due to the increasing needs and increasing pressure to act.

An initial analysis, using a logistic regression and clustering errors by country can confirm some of these assumptions. Table 2 summarizes the results regarding the factors of the demand structure.

**Table 2.** Mention of Rural Youth Education and the demand structure.

| Variables | (1) | (2) | (3) | (4) |
|---|---|---|---|---|
| Rural youth aged 15–19 [1] | 0.0254 ** | | | |
| | (0.0118) | | | |
| Rural NEETs | | 0.0316 | | |
| | | (0.0197) | | |
| Rural early school leavers | | | −0.00866 | |
| | | | (0.0247) | |
| High rural youth education (rural youth 15–24, ISCED 5–8) | | | | −0.0573 * |
| | | | | (0.0303) |
| Constant | −2.139 *** | −1.628 *** | −1.133 *** | −0.814 *** |
| | (0.553) | (0.384) | (0.352) | (0.267) |
| Observations | 523 | 523 | 523 | 472 |
| *AIC* | 548.8586 | 554.4736 | 559.5648 | 519.9399 |
| *BIC* | 557.3778 | 562.9928 | 568.0839 | 528.2538 |

Source: Calculated from own dataset. Representation of coefficients. Notes: dichotomous dependent variable mention of rural youth education (yes/no). [1] The result for 20–24 year-olds is similar, but only significant at a 10% level. Robust standard errors in parentheses (clustered by country). *** $p < 0.01$, ** $p < 0.05$, * $p < 0.1$.

This shows that a high share of rural youth aged 15–19 (and 20–24) has a significantly positive effect on the likelihood of paying attention to rural youth education in policies. As expected, when the need to promote education is less pressing because many rural youths have attained a high education level, the probability of countries including rural education in their policies decreases.

However, neither high rates of rural NEETs nor rural early-school-leaving rates have a significant influence. Figure 4 also supports these findings. While the majority of countries with policies mentioning the education of rural youth decreased their rural NEET and early-school-leaving rates, in countries such as Romania or Slovenia, the reference to rural youth did not have a relevant effect. Besides this, countries without policies addressing rural youth, for example, Luxembourg or Portugal, showed success in tackling early school leaving and rural NEETs. This raises the question whether such programs are effective and to what degree they can reach the right target groups.

Thus, a conceivable explanation could be that a high percentage of youth living in rural areas signals a need for countries to promote education for this group. However, this effect could reverse when rural youth attain high education levels in a country. Following this, a government might not see the need to address rural education explicitly.

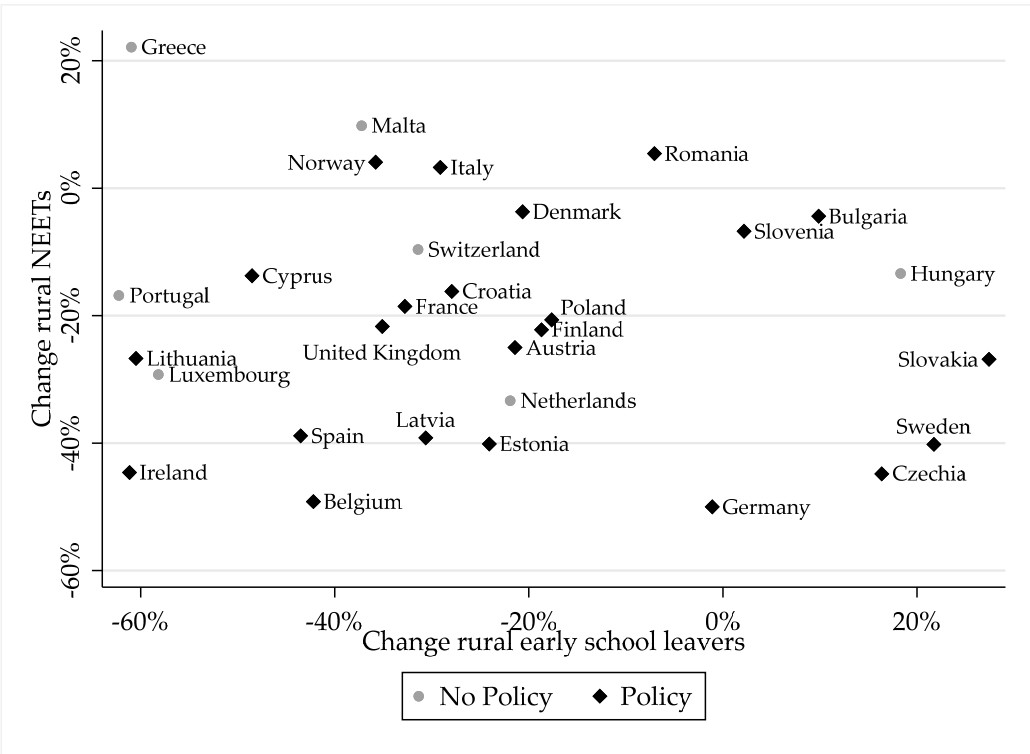

**Figure 4.** Relative change in percentage points of early school leavers (18–24 years) and NEETs (15–24 years) in rural areas 2010–2019 and policies on the education of rural youth. Source: Eurostat 2020, [edat_lfse_29] and [edat_lfse_30], own calculations.

These findings pose as a stress test for the functionalist argument, which further research can analyze in more depth. However, other theoretical factors not included in this analysis might be even more powerful to explain the conditions under which countries consider rural youth. Among these could be the historical background of a country, path dependencies, and party political aspirations, as well as the involvement of civil society, especially in terms of non-formal education. The causal mechanism cannot be identified within the research framework and neither can we evaluate the effectiveness of policies. This opens up more opportunities for future studies.

## 7. Discussion and Conclusions

Although education is the foundation for a successful pathway to employment, the EU has so far focused greatly on youth unemployment while somewhat neglecting the aspect of education. The present analyses thereby presents that, within the EU 2020 strategy, most rural policy ignores youth, and most youth policies neglect the rural dimension. This is a conclusion already reached by existing research [35] on the previous EU policy period, thus underlining a prolonged period of ignoring the needs of rural youth by the EU.

The situation of rural youth in the past ten years has been equally bad, as represented by varying proportions of youth living in rural areas (between countries) and the clear recognition that only 19 of 31 countries achieved the EU target of an early-school-leaving rate below 10% in rural areas. Further, none of the countries achieved the EU goal regarding tertiary education.

Inquiring whether and in what form (sub-)national policies consider the situation of rural youth, the analysis presented here revealed a mixed picture. 19 of 31 countries mention rural youth education somewhere within their national policies; some countries instead target it at regional levels and seven countries do not address it at all. Overall, no country presented a comprehensive strategy targeting rural youth education. If countries target the issue within their policies, the measures taken vary greatly. They range from

concrete measures like financial aid, to plans for restructuring the educational sector and general calls for improvement without naming concrete steps.

This analysis was complemented by an analysis of factors expected to increase the probability of mentioning rural youth education in policies. As expected, with an increasing amount of rural youth, the probability of mentions of rural youth education in policies increases also. Surprisingly, an increased problem pressure (in terms of rural NEET rates and rural early-school-leaving rates) does not increase the probability of mentions.

The findings once again highlight the need for policy change. Existing research indicates that such reforms should neither be carried out by individual policy areas acting alone nor by exclusively short-term measures. For example, in the context of Bulgaria, Petrov [36] calls for a bouquet of changes ranging from changes in tax policy to the introduction of an entirely new regional rural development policy. This demand is underlined by Burusic and colleagues [37] for the Croatian context, as in their view, solo efforts, for example by the Ministry of Education, will not lead to the goal. What must be avoided, as comprehensive studies on individual countries show, is that the situation of NEETs is often forgotten, even though there is a plethora of projects in the case of Spain [38], for example, in the context of promoting the return of young adults from abroad.

Finally, how can the situation of youth in rural areas be improved in the future? Adolescence consists of managing risk and well-being, and in doing so it is not only a task for young people themselves, but also for the institutions that form the structures within which young people must act. In this transition phase, young adults should be able to count on the support of institutions, although the present analysis shows that young adults in rural areas mostly do not have this. Although the goals of rural development potentially collide with the needs of young adults, particularly when advancement through education leads to an exit from the limited opportunities of local rural labor markets [39], resolving this dilemma requires both institutional support for "leaving" rural areas and opportunities for later return and long-term retention [35]. Young adults themselves have various ideas for improving their situation. Examples would be "promoting cooperation between urban and rural educational institutions, for instance through school exchanges, encouraging universities to establish campuses in rural areas, or scholarships for rural students" [3]. It is to be hoped that these demands and ideas will be incorporated into the implementation of further EU strategies.

The present study is not without weaknesses in this regard. Two points in particular should be mentioned. First, the local level is not considered in the chosen format. It would be conceivable to have locally based strategies for the promotion of education in rural areas, which would start directly where the problem lies (geographically speaking). Second, the present study design does not allow for any causal statements. In future research, it would be useful to measure the exact causal relationships between different policies and their effects on young adults in rural areas. In terms of policy learning, this would allow for a deeper understanding of the mechanisms and, in terms of evidence-based policymaking, further improvement in the future. In this regard, another conceivable methodological approach would be to focus more on the specifics of the OMC. In addition to a quantitative approach, the method of discourse analysis would also be conceivable, as would a more in-depth examination of individual countries [40].

The importance of the present analysis is nevertheless made clear by the fact that the great importance of rural youth finally found entrance into the current EU Youth Strategy (2019–2027), which under the module "Moving rural youth forward" addresses the specific needs of young adults in rural areas. Thus, the present analysis provides an important starting point for a broader consideration of policymaking in terms of advancing the situation of young adults in rural areas. Future research should address what changes are associated with the current EU Youth Strategy (2019–2027), which now also explicitly focuses on rural youth. This study provides the necessary point of comparison with the situation "before" the EU Youth Strategy (2019–2027).

**Author Contributions:** Conceptualization, J.W.; Data collection, C.H.-F.; Data Analysis, J.W. and C.H.-F.; Writing and Editing, J.W. and C.H.-F. All authors have read and agreed to the published version of the manuscript.

**Funding:** This research received no external funding.

**Data Availability Statement:** The policy dataset created for this study can be made available upon request from the authors. The data for the regression are publicly available; related sources can be found in Appendix A.

**Acknowledgments:** The authors would like to thank the reviewers for their helpful suggestions and comments. The authors gratefully acknowledge the comments from Jale Tosun and Nicole Schmidt on earlier versions of this paper.

**Conflicts of Interest:** The authors declare no conflict of interest.

## Appendix A

*Appendix A.1. Sources of the Variables*

| Name of Variable | Description | Source | Link |
|---|---|---|---|
| *Dependent variable* | | | |
| Rural Education Mentioned | Education of rural youth mentioned. Yes (1) if rural youth education was explicitly mentioned as a goal; no (0) if only indirect referenced, no specification of age group or only description of situation of rural youth without any goals | see below | see below |
| Education Strategy | National and subnational Education Strategies | EU | https://eurydice.eacea.ec.europa.eu/national-education-systems (accessed on 16 October 2020) |
| | | OECD | https://www.oecd.org/education/reformsfinder.htm (accessed on 16 October 2020) |
| | | UNESCO | https://planipolis.iiep.unesco.org/en (accessed on October 2020) |
| Youth Strategy | National and subnational Youth Strategies | EU | https://eacea.ec.europa.eu/national-policies/en/youthwiki/countries (accessed on 19 October 2020) |
| | | Youth Policy Labs | https://www.youthpolicy.org/nationalyouthpolicies/ (accessed on 19 October 2020) |
| Rural Development Program | National and subnational Rural Development Programmes | EU | https://ec.europa.eu/info/food-farming-fisheries/key-policies/common-agricultural-policy/rural-development/country_en (accessed on 16 October 2020) |
| | | European Network for Rural Development | https://enrd.ec.europa.eu/home-page_en (accessed on 19 October 2020) |
| | | OECD | https://www.oecd-ilibrary.org/urban-rural-and-regional-development/oecd-rural-policy-reviews_19909284 (accessed on 16 October 2020) |
| Share of policies mentioning rural youth education | Education of rural youth mentioned as a proportion (%) of the total number of policies | own calculations based on original dataset | |
| Centralized education | Centralized (1)/decentralized education system (0) | EU | https://eurydice.eacea.ec.europa.eu/national-education-systems (accessed on 16 October 2020) |

| Name of Variable | Description | Source | Link |
|---|---|---|---|
| | *Independent variables* | | |
| Rural Youth 15–19 | Share of rural youth population aged 15–19 (% of total youth population) | Eurostat [lfst_r_pgauwsc], own calculations | https://ec.europa.eu/eurostat/databrowser/view/LFST_R_PGAUWSC/default/table?lang=en (accessed on 2 November 2020) |
| Rural Youth 20–24 | Share of rural youth aged 20–24 (% of total youth population) | Eurostat [lfst_r_pgauwsc], own calculations | https://ec.europa.eu/eurostat/databrowser/view/LFST_R_PGAUWSC/default/table?lang=en (accessed on 2 November 2020) |
| Rural NEETs | Share of rural youth neither in employment nor in education and training aged 15–24 (%) | Eurostat [edat_lfse_29] | https://appsso.eurostat.ec.europa.eu/nui/show.do?dataset=edat_lfse_29&lang=en (accessed on 2 November 2020) |
| Rural early school leavers | Share of rural early school leavers aged 18–24 (%) | Eurostat [edat_lfse_30] | https://ec.europa.eu/eurostat/databrowser/view/EDAT_LFSE_30__custom_514582/default/table (accessed on 2 November 2020) |
| High rural youth education | Share of rural youth aged 15–24 with high educational attainment (ISCED 5–8) | Eurostat [edat_lfs_9913] | http://appsso.eurostat.ec.europa.eu/nui/show.do?dataset=edat_lfs_9913&lang=en (accesed on 2 November 2020) |

*Appendix A.2. Summary Statistics*

| Variable | Obs | Mean | Std. Dev. | Min | Max |
|---|---|---|---|---|---|
| Rural Youth 15–19 | 523 | 33.923 | 14.011 | 5.26 | 63.53 |
| Rural Youth 20–24 | 523 | 28.926 | 13.004 | 5.31 | 61.8 |
| Rural NEETs | 523 | 11.663 | 7.344 | 3.9 | 34.8 |
| Rural early school leavers | 523 | 13.002 | 8.255 | 2.4 | 34.5 |
| High rural youth education | 472 | 6.183 | 3.767 | 1 | 18.9 |

*Appendix A.3. Share of Policies Mentioning Education of Rural Youth as a Percentage of the Policies Analyzed for This Country*

A look at the share of policies taking rural education into consideration reveals that only in one country are rural youth addressed in all policies analyzed: Romania. However, the majority of European countries do not put such an emphasis on this aspect and only regard rural youth in a few (less than 40%) of their policies.

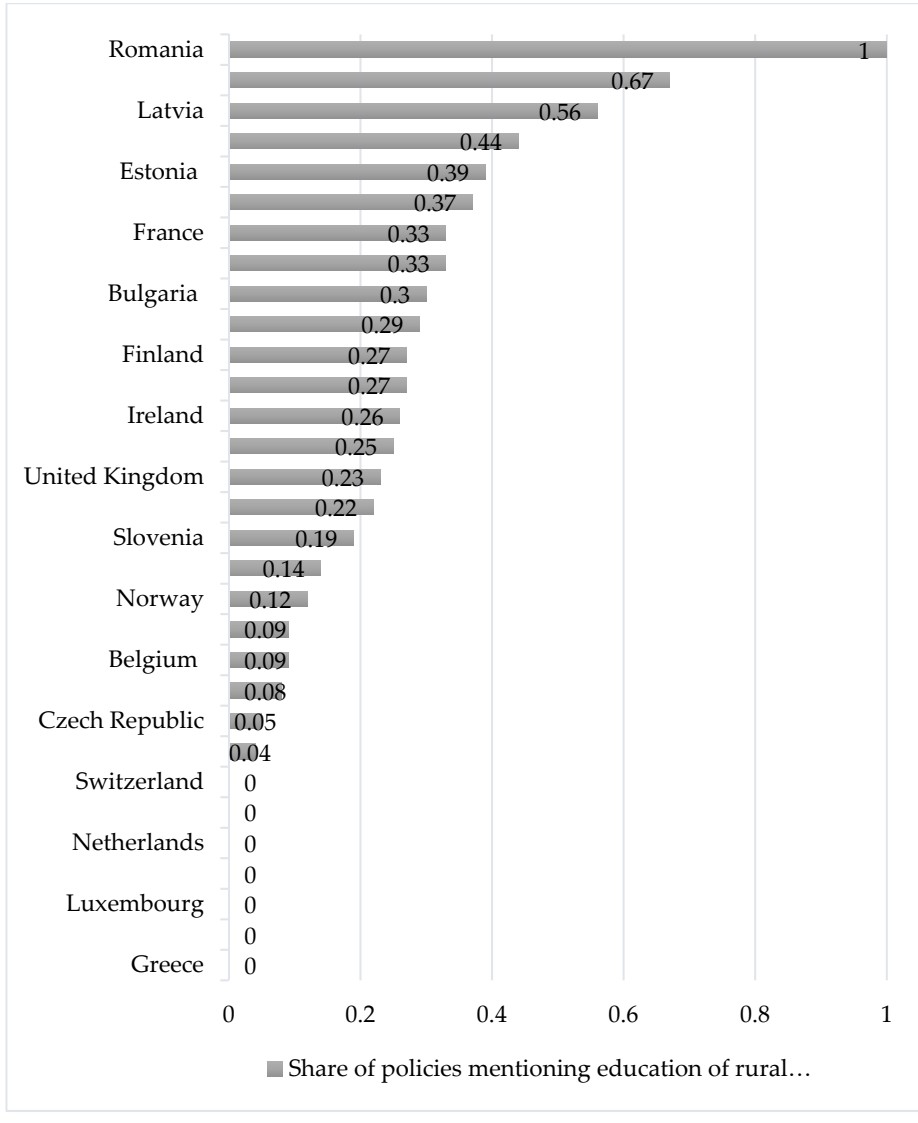

Source: Own dataset.

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
