# Peer review of "The More Rural the Less Educated? An Analysis of National Policy Strategies for Enhancing Young Adults’ Participation in Formal and Informal Training in European Rural Areas"

_2673-995X, doi:10.3390/youth2030030_

Round 1

Reviewer 1 Report

The article describes research of great interest concerning the educational policies of rural young adults in European countries. The authors justify the need for the study adequately, due to the high number of these individuals and the problems associated with the phenomena of NEETs and early school leavers.

It is recommended to rewrite the abstract according to the recommendations of the journal:

The abstract should be a total of about 200 words maximum. The abstract should be a single paragraph and should follow the style of structured abstracts, but without headings: 1) Background: Place the question addressed in a broad context and highlight the purpose of the study; 2) Methods: Describe briefly the main methods or treatments applied. Include any relevant preregistration numbers, and species and strains of any animals used. 3) Results: Summarize the article's main findings; and 4) Conclusion: Indicate the main conclusions or interpretations.

The nature of the research is qualitative. It would be necessary to further detail the research methodology carried out, indicating how the coding process took place, how many people carried it out and what tools or software were used if any.

Table 2 appears in the discussion section and should appear in the results section.

The authors are encouraged to expand the discussion section by considering the academic literature and similar research that has been carried out in the European countries under study. Similarly, it is necessary to include in the discussion and conclusions section future perspectives in a clearer way.

It would also be appropriate to check to what extent countries that mention rural youth in their policies have experienced a decrease in the number of NEETs and early school leavers in this population.  To what extent are these policies effective or not?

Reviewer 2 Report

Thank you for the opportunity to review this manuscript. Authors have done great job tapping an important issue concerning an important population. While the share of youth which are neither in employment nor in education or training in the youth population is considered relatively new variable of interest in research, international organizations and those interested in health indicators are giving it increasing attention. This is a timely work that can have significant contribution to the literature and can inform policy makers. 

One issue that I believe needs refinement is related to the discussion section, which seemed to end prematurely. While informative by themselves, the results were primarily descriptive.  The discussion did not provide sufficient critique for the revealed findings. This is possibly why the recommendations were general.

Minor issue: The acronym NEET needs to be stated fully first time mentioned.

Thank you.
